# Locality Sensitive Sparse Encoding for Learning World Models Online

**Zichen Liu**[†‡]    **Chao Du**[†]    **Wee Sun Lee**[‡]    **Min Lin**[†]

[†]Sea AI Lab    [‡]National University of Singapore

## Abstract

Acquiring an accurate world model *online* for model-based reinforcement learning (MBRL) is challenging due to data nonstationarity, which typically causes catastrophic forgetting for neural networks (NNs). From the online learning perspective, a Follow-The-Leader (FTL) world model is desirable, which optimally fits all previous experiences at each round. Unfortunately, NN-based models need re-training on all accumulated data at every interaction step to achieve FTL, which is computationally expensive for lifelong agents. In this paper, we revisit models that can achieve FTL with incremental updates. Specifically, our world model is a linear regression model supported by nonlinear random features. The linear part ensures efficient FTL update while the nonlinear random feature empowers the fitting of complex environments. To best trade off model capacity and computation efficiency, we introduce a locality sensitive sparse encoding, which allows us to conduct efficient sparse updates even with very high dimensional nonlinear features. We validate the representation power of our encoding and verify that it allows efficient online learning under data covariate shift. We also show, in the Dyna MBRL setting, that our world models learned online using a *single pass* of trajectory data either surpass or match the performance of deep world models trained with replay and other continual learning methods.

## 1 Introduction

World models have been demonstrated to enable sample-efficient model-based reinforcement learning (MBRL) (Sutton, 1990; Janner et al., 2019; Kaiser et al., 2020; Schrittwieser et al., 2020), and are deemed as a critical component for next-generation intelligent agents (Sutton et al., 2022; LeCun, 2022). Unfortunately, learning accurate world models in an incremental online manner is challenging, because the data generated from agent-environment interaction is nonstationarily distributed, due to the continually changing policy, state visitation, and even environment dynamics. For neural network (NN) based world models, the nonstationarity could lead to *catastrophic forgetting* (McCloskey & Cohen, 1989; French, 1999), making models inaccurate at recently under-visited regions (as illustrated in the top row of Figure 1). Planning with inaccurate models is detrimental as the model errors can compound due to rollouts, resulting in misleading agent updates (Talvitie, 2017; Jafferjee et al., 2020; Wang et al., 2021; Liu et al., 2023). To attain a world model of good accuracy over the observed data coverage, NN-based methods often maintain all collected experiences from the start of the environment interaction and perform periodic re-training, possibly at every step, resulting in a growing computation cost for lifelong agents.

In this paper, we aim to develop a world model, in the Dyna MBRL architecture (Sutton, 1990; 1991), that learns incrementally without forgetting prior knowledge about the environment. We notice that re-training NN-based world models every step on previous observations till convergence resembles the concept of Follow-The-Leader (FTL) in online learning (Shalev-Shwartz et al., 2012). While re-training the NN on all previous data is prohibitively expensive, especially under the streaming setting in RL, specific types of models studied in online learning can achieve incremental FTL with only a constant computation cost. To this end, we revisit the classic idea of learning a linear regressor on top of non-linear random features. The loss function of such models is quadratic with respect to the parameters, satisfying the online learning requirement. It may seem a retrograde choice given all the success of deep learning, and the findings that a shallow NN needs to be exponentially large to match the capability of a deep one (Eldan & Shamir, 2016; Telgarsky, 2016).

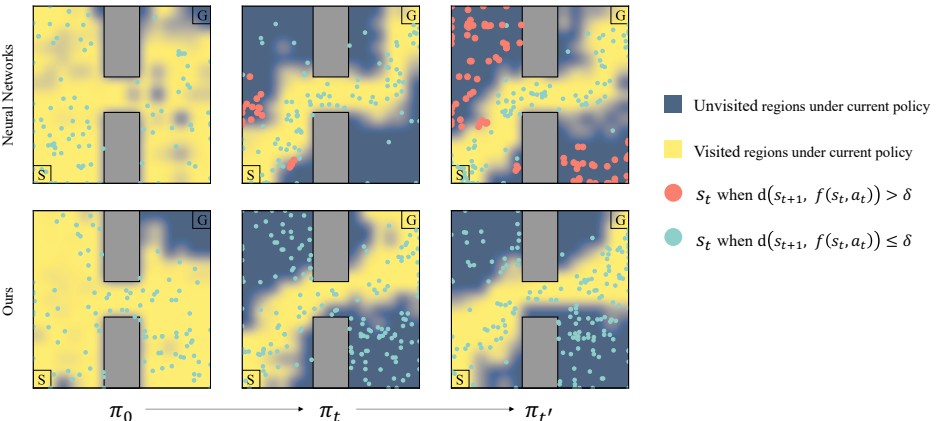

**Figure 1:** This Gridworld environment requires the agent to navigate from the start position ("S") to the goal location ("G") with shortest path. The tabular Q-learning agent starts from a random policy $\pi_0$ and improves to get better policies $\pi_t \cdots \pi_{t'}$, leading to narrower state visitation towards the optimal trajectories (*the yellow regions*). Due to such distributional shift, the NN-based model (*top*) may forget recently under-visited regions, even though it has explored there before. The red circles indicate erroneous predictions where the Euclidean distance between the ground truth next state and the prediction is greater than a threshold ($\delta = 0.05$). In contrast, our method (*bottom*) learns online, and at each step incrementally computes the optimal solution over all accumulated data, thus is resilient to forgetting.

Nevertheless, we believe it is worth exploring under the RL context for three reasons: (1) In RL, data is streamed online and highly non-stationary, advocating for models capable of incremental FTL. (2) Many continuous control problems have a moderate number of dimensions that could fall within the capability of shallow models (Rajeswaran et al., 2017). (3) Sparse features can be employed to further enlarge the model capacity without extra computation cost (Knoll & de Freitas, 2012).

Inspired by Knoll & de Freitas (2012), we propose an expressive *sparse non-linear feature* representation which we call locality sensitive sparse encoding. Our encoder generates high-dimensional sparse features with random projection and soft binning. Exploiting the sparsity, we further develop an efficient algorithm for online model learning, which only updates a small subset of weights while *continually tracking a solution* to the FTL objective. World models learned online with our method are resilient to forgetting compared to those based on NNs (see the bottom row of Figure 1 for intuition). We empirically validate the representational advantage of our encoding over other non-linear features, and demonstrate that our method outperforms NNs in the online supervised learning setting (Orabona, 2019; Hoi et al., 2021) as well as the model-based reinforcement learning setting (Sutton, 1990) with models learned online.

## 2 PRELIMINARIES

In this section, we first recap the protocol of online learning and introduce the Follow-The-Leader strategy. Then we revisit Dyna, a classic MBRL architecture, where we apply our method to learn world models online.

### 2.1 ONLINE LEARNING

Online learning refers to a learning paradigm where the learner needs to make a sequence of accurate predictions given knowledge about the correct answers for all prior questions (Shalev-Shwartz et al., 2012). Formally, at round $t$, the online learner is given a question $\mathbf{x}_t \in \mathcal{X}$ and asked to provide an answer to it, which we denote as $h_t(\mathbf{x}_t)$, letting $h_t \in \mathcal{H} : \mathcal{X} \to \mathcal{Y}$ be a model in the hypothesis class built by the learner. After predicting the answer, the learner will be revealed the ground truth $\mathbf{y}_t \in \mathcal{Y}$ and suffers a loss $\ell(h_t(\mathbf{x}_t), \mathbf{y}_t)$. The learner's goal is to adjust its model to achieve the lowest possible regret relative to $\mathcal{H}$ defined as $\text{Regret}_T(\mathcal{H}) = \max_{h^* \in \mathcal{H}} \left( \sum_{t=1}^{T} \ell(h_t(\mathbf{x}_t), \mathbf{y}_t) - \sum_{t=1}^{T} \ell(h^*(\mathbf{x}_t), \mathbf{y}_t) \right)$. When the input is a convex set $\mathcal{S}$, the prediction is a vector $\mathbf{w}_t \in \mathcal{S}$, and the loss $\ell(\mathbf{w}_t, \mathbf{y}_t)$ is convex, the problem is cast as *online convex optimization*. Absorbing the target into the loss, $\ell_t(\mathbf{w}_t) = \ell(\mathbf{w}_t, \mathbf{y}_t)$, the regret with respect to a

competing hypothesis (a vector $\mathbf{u}$) is then defined as $\text{Regret}_T(\mathbf{u}) = \sum_{t=1}^T \ell_t(\mathbf{w}_t) - \sum_{t=1}^T \ell_t(\mathbf{u})$. One intuitive strategy is for the learner to predict a vector $\mathbf{w}_t$ at any online round such that it achieves minimal loss over all past rounds. This strategy is usually referred to as Follow-The-Leader (FTL):

$$\forall t, \ \mathbf{w}_t = \arg\min_{\mathbf{w} \in \mathcal{S}} \sum_{i=1}^{t-1} \ell_i(\mathbf{w}). \tag{1}$$

## 2.2 MODEL-BASED REINFORCEMENT LEARNING WITH DYNA

Reinforcement learning problems are usually formulated with the standard Markov Decision Process (MDP) $\mathcal{M} = \{\mathcal{S}, \mathcal{A}, P, R, \gamma, P_0\}$, where $\mathcal{S}$ and $\mathcal{A}$ denote the state and action spaces, $P(\mathbf{s}'|\mathbf{s}, \mathbf{a})$ the Markovian transition dynamics, $R(\mathbf{s}, \mathbf{a}, \mathbf{s}')$ the reward function, $\gamma \in (0, 1)$ the discount factor, and $P_0$ the initial state distribution. The goal of RL is to learn the optimal policy that maximizes the discounted cumulative reward: $\pi^* = \arg\max_\pi \mathbb{E}_{\pi,P} \left[ \sum_{t=0}^\infty \gamma^t R(\mathbf{s}_t, \mathbf{a}_t, \mathbf{s}_{t+1}) \mid \mathbf{s}_0 = \mathbf{s} \sim P_0 \right]$. Model-based RL solves the optimal policy with a learned world model. Existing NN-based MBRL methods (Schrittwieser et al., 2020; Hansen et al., 2022; Hafner et al., 2023) have shown state-of-the-art performance on various domains but rely heavily on techniques to make data more stationary, such as maintaining a large replay buffer or periodically updating the target network. With these components removed, they would fail to work. When they fail, however, it is unclear how much is due to the forgetting in policies, value functions, or world models because all components are entangled and trained end-to-end in these methods. Therefore, we focus on Dyna, which learns a *world model* (or simply *model* in RL literature) alongside learning the *base agent*[1] in a decoupled manner.

Figure 2 illustrates the Dyna architecture that we employ in this paper to study online model learning. The base agent will act in the environment and collect *environment experiences*, which can be used for learning the world model to mimic the environment's behavior. With such a model learned, the agent then simulate with it to synthesize *model experiences* for planning updates. *Search control* defines how the agent queries the model, and some commonly used ones include predecessor (Moore & Atkeson, 1993), on-policy (Janner et al., 2019) and hill-climbing (Pan et al., 2019). Next, we define how to learn and plan with the model.

**Figure 2:** The Dyna architecture.

**Model learning.** We first formulate how to learn the dynamics model $m(\mathbf{s}'|\mathbf{s}, \mathbf{a})$ with environment experiences. Instead of directly predicting the next state, we model the dynamics as an integrator which estimates the change of the successive states $\hat{\mathbf{s}}' = \mathbf{s} + \Delta t \cdot \hat{m}(\Delta \mathbf{s}|\mathbf{s}, \mathbf{a})$, where $\Delta t$ is the discrete time step. This technique enables more stable predictions and is widely adopted (Chua et al., 2018; Janner et al., 2019; Yu et al., 2020). The dynamics can either be modeled as a stochastic or a deterministic function, with the former predicting the state-dependent variance besides the mean to capture aleatoric uncertainty in the environment (Chua et al., 2018). We focus on learning a deterministic dynamics model since we are mainly interested in environments without aleatoric uncertainty. As shown in Lutter et al. (2021), deterministic dynamics models can obtain comparable performance with stochastic counterparts. Even when the environment exhibits stochasticity, generating rollouts with fixed variance is also as effective as learned variance (Nagabandi et al., 2020). Therefore, dynamics model learning is a regression problem with the objective

$$\mathbb{E}_{(\mathbf{s}, \mathbf{a}, \mathbf{s}') \sim \mathcal{D}_{\text{env}}} \|\hat{m}(\mathbf{s}, \mathbf{a}) - \Delta \mathbf{s}\|_2^2, \tag{2}$$

where $\mathcal{D}_{\text{env}}$ is the environment experiences and $\Delta \mathbf{s} = \mathbf{s}' - \mathbf{s}$. The reward model learning follows a similar regression objective and we will omit it throughout this paper for brevity.

**Model planning.** In Dyna-style algorithms, the learned model is meant for the agent to interact with, as a substitution of the real environment. When the agent "plans" with the model, it queries the model at $(\tilde{\mathbf{s}}, \tilde{\mathbf{a}}) \in \mathcal{S} \times \mathcal{A}$ to synthesize model experiences $\mathcal{D}_m = (\tilde{\mathbf{s}}, \tilde{\mathbf{a}}, \hat{r}, \hat{\mathbf{s}}')$, which serves as a data source for agent learning. There are various ways to determine how to query the model. For example,

---

[1]We use the term *base agent* to refer to other components than the *model*, e.g., the value function and policy.

the classic Dyna-Q (Sutton, 1990) conducts one-step simulations on state-action pairs uniformly sampled from prior experiences. More recently, MBPO (Janner et al., 2019) generalizes the one-step simulation to short-horizon on-policy rollout branched from observed states and provides theoretical analysis for the monotonic improvement of model-based policy optimization.

## 3  LEARNING WORLD MODELS ONLINE

The agent-environment interaction generates a temporally correlated data stream $(\mathbf{s}_t, \mathbf{a}_t, r_t, \mathbf{s}_{t+1})_t$. At each time step, the world model observes $\mathbf{x}_t = [\mathbf{s}_t, \mathbf{a}_t] \in \mathbb{R}^{S+A}$, predicts the next state and suffers a quadratic loss $\ell_t(\hat{m}_t) = \|\hat{m}_t(\mathbf{x}_t) - \mathbf{y}_t\|_2^2$, where $S$ and $A$ are the dimensionalities of state and action spaces, $\mathbf{y}_t = (\mathbf{s}_{t+1} - \mathbf{s}_t) \in \mathbb{R}^S$ is the state difference target. In most model-based RL algorithms built with neural networks, the model learning is conducted in a supervised offline manner, with periodic re-training over the whole dataset (see the objective in Equation 2). Such methods achieve FTL for NNs but are inefficient due to the growing size of the dataset. We develop in this section an online method for learning world models with efficient incremental update on every single environment step.

### 3.1  ONLINE FOLLOW-THE-LEADER MODEL LEARNING

Our world model $\hat{m}(\mathbf{x}) = \phi(\mathbf{x})^\top \mathbf{W}$ employs a linear function approximation with weights $\mathbf{W} \in \mathbb{R}^{D \times S}$ on an expressive and sparse non-linear feature $\phi(\mathbf{x}) \in \mathbb{R}^D$. The FTL objective in Equation 1 thus converts to a least squares loss per time step:

$$\forall t, \ \mathbf{W}_t = \underset{\mathbf{W} \in \mathbb{R}^{D \times S}}{\arg\min} \|\mathbf{\Phi}_{t-1} \mathbf{W} - \mathbf{Y}_{t-1}\|_F^2, \tag{3}$$

where $\mathbf{\Phi}_\tau = [\phi(\mathbf{x}_1), \ldots, \phi(\mathbf{x}_\tau)]^\top \in \mathbb{R}^{\tau \times D}$ denotes non-linear features for observations accumulated until time step $\tau$, $\mathbf{Y}_\tau = [\mathbf{y}_1, \ldots, \mathbf{y}_\tau]^\top \in \mathbb{R}^{\tau \times S}$ is the corresponding targets, and $\| \cdot \|_F$ denotes the Frobenius norm. We can expect our world model with the weights satisfying Equation 3 to have good regret, as the regret of FTL for online linear regression problems with quadratic loss is $\text{Regret}_T(\mathbf{W}) = \mathcal{O}(\log T)$ (Shalev-Shwartz et al., 2012; Gaillard et al., 2019). As a consequence, our online learned world model will be immune to forgetting, rendering it suitable for lifelong agents.

In the following sections, we present how to attain a feature representation that is expressive enough for modeling complex transition dynamics (Section 3.2), as well as how its sparsity helps to achieve efficient incremental update (Section 3.3).

### 3.2  FEATURE REPRESENTATION

We construct high-dimensional sparse features using random projection followed by soft binning. The input vector $\mathbf{x}_t \in \mathbb{R}^{S+A}$ is first projected into feature space $\sigma : \mathbf{x} \to \mathbf{P}\mathbf{x}$, where $\mathbf{P} \in \mathbb{R}^{d \times (S+A)}$ is a random projection matrix sampled from a multivariate Gaussian distribution so that the transformation approximately preserves similarity in the original space (Johnson & Lindenstrauss, 1984). Afterwards, each element of $\sigma(\mathbf{x}_t)$ is binned in a soft manner by locating its neighboring edges as *indices* and computing the distances between all edges as *values*. We denote the binning operation as $b : \mathbb{R}^d \to \mathbb{R}^D$, where $D \gg d$. Compared to naive binning which produces binary one-hot vectors and loses precision, ours has greater discriminative power by generating multi-hot real-valued representations.

We provide the following example to illustrate. Assume we are binning $\sigma_1(\mathbf{x}_t) = 1.7$ into a 1-d grid with edges $[0, 1, 2, 3]$. The naive binning would generate a vector $[0, 1, 0]$ to indicate the value falls into the central bin. In comparison, the soft binning first locates the neighbors 1 and 2 as indices, and computes distances to the neighbors as values, thus forming a vector $[0, 0.7, 0.3, 0]$. Our soft binning can discriminate two inputs falling inside the same bin, such as $\sigma_1(\mathbf{x}_t) = 1.7$ and $\sigma_1(\mathbf{x}_t) = 1.2$, while naive binning fails to do so.

Altogether, our feature encoder $\phi = b \circ \sigma$ resembles Locality Sensitive Hashing (LSH) with random projections (Charikar, 2002) but is more expressive thanks to soft binning. We term our feature representation as **Lo**cality **s**ensitive **s**parse **e**ncoding, hence **Losse**. We depict the overall Losse process in Figure 3.

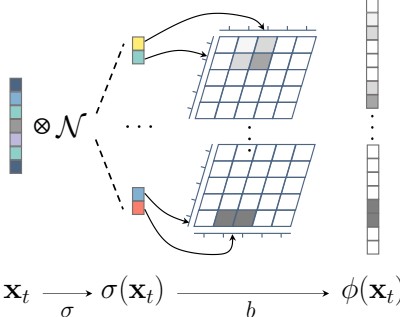

$$\mathbf{x}_t \xrightarrow{\sigma} \sigma(\mathbf{x}_t) \xrightarrow{b} \phi(\mathbf{x}_t)$$

**Figure 3:** Locality sensitive sparse encoding. $\sigma(\cdot)$ projects input vectors into a random feature space, and $b(\cdot)$ softly bins $\sigma(\mathbf{x}_t)$ into multiple $\rho$-dimensional grids, which are flattened and stacked into a high-dimensional sparse encoding $\phi(\mathbf{x}_t)$.

We note that the raw output of $b$ can be multi-dimensional before flattening (e.g. 2-dimensional in Figure 3), and we use $\rho$ to denote its dimensionality. A $\rho$-dimensional grid has $\lambda$ evenly spaced bins along each axis, thus producing a highly sparse feature vector of length $\lambda^\rho$ per grid. Finally, we can stack $\kappa$ grids with different random projection directions to get diverse features. Losse has guaranteed sparsity as stated in Remark 3.1.

**Remark 3.1** (Sparsity guarantee of Losse). *For any $\rho, \lambda, \kappa \in \mathcal{Z}_{>0}$, and any input vector $\mathbf{x}$, $\phi(\mathbf{x})$ outputs a vector whose number of nonzero entries $\|\phi(\mathbf{x})\|_0$ satisfies $\|\phi(\mathbf{x})\|_0 \leq \kappa 2^\rho$, and the proportion of nonzero entries in $\phi(\mathbf{x})$ is at most $\left(\frac{2}{\lambda}\right)^\rho$.*

To give a concrete example, if we set $\rho = 3$ and $\lambda = 10$, then $\|\phi(\mathbf{x})\|_0$ is at most $8\kappa$ while the dimension of $\phi(\mathbf{x})$ can be as high as $1000\kappa$. The high-dimensional feature brings us more fitting capacity, while the bounded sparsity permits efficient model updates with constant overheads, as we will show in the following section.

### 3.3 EFFICIENT SPARSE INCREMENTAL UPDATE

A no-regret world model after $t$ steps of interaction can be updated online by computing the weights $\mathbf{W}_{t+1}$ (Equation 3), which can be solved in *closed form* using the normal equation: $\mathbf{W}_{t+1} = \mathbf{A}_t^{-1}\mathbf{B}_t$, where $\mathbf{A}_t = \mathbf{\Phi}_t^\top\mathbf{\Phi}_t$ and $\mathbf{B}_t = \mathbf{\Phi}_t^\top\mathbf{Y}_t$ are two memory matrices. Hence, we obtain a simple algorithm (Algorithm 1) to learn the model online at every time step.

---

**Algorithm 1** Online model learning

1: $t = 0, \mathbf{A}_0 \in \mathbb{R}^{D \times D} = \mathbf{0}, \mathbf{B}_0 \in \mathbb{R}^{D \times S} = \mathbf{0}$
2: **while** True **do**
3:     $t \leftarrow t + 1$
4:     $\mathbf{A}_t \leftarrow \mathbf{A}_{t-1} + \phi(\mathbf{x}_t)\phi(\mathbf{x}_t)^\top$
5:     $\mathbf{B}_t \leftarrow \mathbf{B}_{t-1} + \phi(\mathbf{x}_t)\mathbf{y}_t^\top$
6:     $\mathbf{W}_{t+1} \leftarrow \mathbf{A}_t^{-1}\mathbf{B}_t$
7: **end while**

---

**Algorithm 2** *Sparse* online model learning

1: $t = 0, \mathbf{A}_0 \in \mathbb{R}^{D \times D} = \mathbf{0}, \mathbf{B}_0 \in \mathbb{R}^{D \times S} = \mathbf{0}$
2: **while** True **do**
3:     $t \leftarrow t + 1$
4:     $s \leftarrow \mathtt{nonzero\_index}(\phi(\mathbf{x}_t))$
5:     $\mathbf{A}_{t,ss} \leftarrow \mathbf{A}_{t-1,ss} + \phi_s(\mathbf{x}_t)\phi_s(\mathbf{x}_t)^\top$
6:     $\mathbf{B}_{t,s} \leftarrow \mathbf{B}_{t-1,s} + \phi_s(\mathbf{x}_t)\mathbf{y}_t^\top$
7:     $\mathbf{W}_{t+1,s} \leftarrow \mathbf{A}_{t,ss}^{-1}(\mathbf{B}_{t,s} - \mathbf{A}_{t,s\bar{s}}\mathbf{W}_{t,\bar{s}})$
8: **end while**

---

Updating the memory matrices $\mathbf{A}_t, \mathbf{B}_t$ is relatively cheap, but computing weights (line 6 of Algorithm 1) involves matrix inversion, which is computationally expensive. Though $\mathbf{W}_{t+1}$ can be recursively updated using Sherman-Morrison formula (Sherman & Morrison, 1950) as done in Least-Squares TD (Bradtke & Barto, 1996), when the feature becomes very high-dimensional to gain expressiveness, it can become inefficient or even impractical.

Fortunately, the incremental updates $\phi(\mathbf{x}_t)\phi(\mathbf{x}_t)^\top$ and $\phi(\mathbf{x}_t)\mathbf{y}_t^\top$ are highly sparse by Losse, so that we can efficiently update a small subset of weights. As shown in Algorithm 2, we first record the indices where the softly binned random features fall into (line 4), which are used to locate those "activated" entries in the current step and update only a small block of the memory matrices (line 5-6). Most importantly, the matrix inversion for updating the weight subset can be conducted on a much smaller sub-matrix efficiently while ensuring optimality (line 7). The cardinality of the index set $s$ is bounded by the number of non-zero features ($\kappa 2^\rho$), giving constant compute cost regardless of the number of data samples. We next derive how our sparse weight update rule solves Equation 3 in a block-wise manner, which yields an optimal fitting for all data including the current one at each update step.

We first re-index the sparse vector such that it is the concatenation of two vectors $\phi(\mathbf{x}) = [\phi_{\bar{s}}(\mathbf{x}), \phi_s(\mathbf{x})]$, where all zero entries go to the former and the latter is densely real-valued. We discard the subscript on time step for notational convenience. Intuitively, the new observation $\mathbf{x}$

will only update a subset of weights associated with its non-zero feature entries, which we denote as $\mathbf{W}_s \in \mathbb{R}^{K \times S}$, where $K$ is the dimension of $\phi_s(\mathbf{x})$. Then $\mathbf{W}_{\overline{s}} \in \mathbb{R}^{(D-K) \times S}$ is the complement weight matrix. Correspondingly, the accumulative memories $\mathbf{A} = \mathbf{\Phi}^\top \mathbf{\Phi}$ and $\mathbf{B} = \mathbf{\Phi}^\top \mathbf{Y}$ can be decomposed as

$$\mathbf{A} = \begin{pmatrix} \mathbf{A}_{\overline{s}\overline{s}} & \mathbf{A}_{\overline{s}s} \\ \mathbf{A}_{s\overline{s}} & \mathbf{A}_{ss} \end{pmatrix}, \quad \mathbf{B} = \begin{pmatrix} \mathbf{B}_{\overline{s}} \\ \mathbf{B}_s \end{pmatrix}. \tag{4}$$

Then the objective in Equation 3 becomes

$$\underset{\mathbf{W}}{\arg\min} \left( \|\mathbf{\Phi W}\|_F^2 + \|\mathbf{Y}\|_F^2 - 2 \langle \mathbf{\Phi W}, \mathbf{Y} \rangle_F \right)$$

$$\Leftrightarrow \underset{\mathbf{W}}{\arg\min} \left[ \mathrm{Tr} \left( \begin{pmatrix} \mathbf{W}_{\overline{s}} \\ \mathbf{W}_s \end{pmatrix}^\top \begin{pmatrix} \mathbf{A}_{\overline{s}\overline{s}} & \mathbf{A}_{\overline{s}s} \\ \mathbf{A}_{s\overline{s}} & \mathbf{A}_{ss} \end{pmatrix} \begin{pmatrix} \mathbf{W}_{\overline{s}} \\ \mathbf{W}_s \end{pmatrix} \right) - 2 \mathrm{Tr} \left( \begin{pmatrix} \mathbf{W}_{\overline{s}} \\ \mathbf{W}_s \end{pmatrix}^\top \begin{pmatrix} \mathbf{B}_{\overline{s}} \\ \mathbf{B}_s \end{pmatrix} \right) \right], \tag{5}$$

where $\langle \cdot, \cdot \rangle_F$ denotes the Frobenius inner product. Treating $\mathbf{W}_{\overline{s}}$ as a constant since it is not affected by current data and optimizing $\mathbf{W}_s$ only gives

$$\underset{\mathbf{W}_s}{\arg\min} \left( \underline{\mathrm{Tr}(\mathbf{W}_{\overline{s}}^\top \mathbf{A}_{\overline{s}\overline{s}} \mathbf{W}_{\overline{s}})} + \mathrm{Tr}(\mathbf{W}_s^\top \mathbf{A}_{ss} \mathbf{W}_s) + 2 \mathrm{Tr}(\mathbf{W}_s^\top \mathbf{A}_{s\overline{s}} \mathbf{W}_{\overline{s}}) \right.$$

$$\left. - \underline{2 \mathrm{Tr}(\mathbf{W}_{\overline{s}}^\top \mathbf{B}_{\overline{s}})} - 2 \mathrm{Tr}(\mathbf{W}_s^\top \mathbf{B}_s) \right). \tag{6}$$

The new objective in Equation 6 is quadratic, so we differentiate with respect to $\mathbf{W}_s$ and set the derivative to zero to obtain the solution of the sub-matrix

$$\mathbf{W}_s = \mathbf{A}_{ss}^{-1} (\mathbf{B}_s - \mathbf{A}_{s\overline{s}} \mathbf{W}_{\overline{s}}), \tag{7}$$

which gives us the sparse update rule in Algorithm 2.

**Concluding remark**. So far, we have presented an efficient algorithm that learns world models online without forgetting. In our algorithm, linear models are exploited so that we can achieve no-regret online learning with Follow-The-Leader. To enlarge the model capacity, we devise a high-dimensional nonlinear random feature encoding, Losse, turning linear models into universal approximators (Huang et al., 2006). Moreover, the sparsity guarantee of Losse permits efficient updates of the shallow but wide model. We name our method ***Losse-FTL***. We note that feature sparsity has been utilized to mitigate catastrophic forgetting for decades (McCloskey & Cohen, 1989; French, 1991; Liu et al., 2019; Lan & Mahmood, 2023). Our work differs from them in that not only does sparsity help reduce feature interference, but the incremental closed-form solution also guarantees the optimal fitting of *all* observed data, thus eliminating forgetting. More discussion on feature sparsity can be found in Appendix A. Another related line of research to ours is analytic class-incremental learning (Zhuang et al., 2022; 2023; 2024), which we discuss in Appendix B due to space constraints. The implementation details of Losse-FTL can be found in Appendix E.

## 4  EMPIRICAL RESULTS ON SUPERVISED LEARNING

In this section, we will showcase empirical results to validate the expressiveness of Losse (Section 4.1) and the online learning capability of Losse-FTL (Section 4.2), both in supervised learning settings.

### 4.1  COMPARING FEATURE REPRESENTATIONS

We compare our locality sensitive sparse encoding with other feature encoding techniques, including Random Fourier Features (Rahimi & Recht, 2007), Random ReLU Features (Sun et al., 2018), and Random Tile Coding, which combines random projection and Tile Coding (Sutton & Barto, 2018). We note that in all these methods including ours, random projections are used to construct high-dimensional features to acquire more representation power. However, they have different properties with regards to the sparsity and feature values, as summarized in Table 1. Losse is designed to be sparse and real-valued. Sparse encoding allows for more efficient incremental update (Algorithm 2), while distance-based real values provide better generalization. Although ReLU also produces sparse features, it does not guarantee the sparsity level, which is dependent on the sign of its outputs.

We test different encoding methods on an image denoising task, where the inputs are MNIST (Deng, 2012) images with Gaussian noise, and the outputs are clean images. The flattened noisy images

are first encoded by different methods to produce feature vectors, on which a linear layer is applied to predict the clean images of the same size. We use mini-batch stochastic gradient descent to optimize the weights of the linear layer until convergence. To keep the online update efficiency similar, we keep the same number of non-zero entries for all encoding methods. The mean squared errors on the test set for different patch sizes are reported in Table 1. We observe that Losse achieves the lowest error across all patch sizes, justifying its representational strength over other non-linear feature encoders. When compared with the NN baseline, which is a strong function approximator but not an efficient online learner, linear models with Losse work better when the patch size is 36 or smaller, suggesting its sufficient capacity for problems with a moderate number of dimensions, such as locomotion or robotics (Todorov et al., 2012; Zhu et al., 2020). See Appendix C.1 for more experimental details.

|  | Sparse | Real-value | 9 | 16 | 25 | 36 | 49 |
|---|---|---|---|---|---|---|---|
| NN | - | - | 0.36 | 0.35 | 0.34 | 0.34 | 0.33 |
| Fourier | ✗ | ✓ | 0.32 | 0.39 | 0.67 | 1.00 | 1.40 |
| ReLU | ✓ | ✓ | 0.35 | 0.34 | 0.37 | 0.39 | 0.43 |
| Tile Code | ✓ | ✗ | 0.36 | 0.39 | 0.51 | 0.61 | 0.73 |
| **Losse** | ✓ | ✓ | **0.28** | **0.29** | **0.31** | **0.34** | **0.40** |

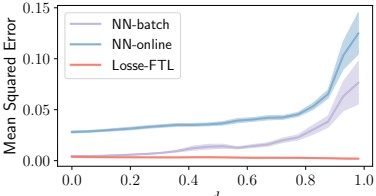

**Table 1:** Properties of different encoding methods and their mean squared errors on the image denoising task on different patch sizes. All numbers are scaled by $10^{-1}$.

**Figure 4:** Mean squared errors on the stream learning task of different correlation levels. Solid lines and shaded areas correspond to the means and stand errors of 30 runs.

## 4.2 ONLINE LEARNING WITH COVARIATE SHIFT

Next, we consider a supervised stream learning setting, where we can precisely control the level of covariate shift in the training data, and test the online learning capability of Losse-FTL against the neural network counterpart. Similar to Pan et al. (2021), we create a synthetic data stream with observations sampled from a non-stationary input distribution. Specifically, the synthetic data is generated according to a Piecewise Random Walk. At each time step, the observation is sampled from a Gaussian $X_t \sim \mathcal{N}(S_t, \beta^2)$, where $\beta$ is fixed and $S_t$ drifts every $\tau$ steps. The drifting follows a Gaussian first order auto-regressive random walk $S_{t+1} = (1-c)S_t + Z_t, \forall t \mod \tau \equiv 0$, where $c \in (0, 1]$ and $Z_t \sim \mathcal{N}(0, \sigma^2)$ with a fixed $\sigma$. For $X_t = x_t$, the target is defined as $y_t = \sin(2\pi x_t^2)$.

As proven in Pan et al. (2021), $X_t$ can share the same equilibrium distribution but possess different levels of temporal correlation if $\beta, c$, and $\sigma$ are properly chosen. It turns out the three scalars can be uniquely determined by a single parameter $d \in [0, 1)$, which we call *correlation level*. When $d = 0$, the generated data recovers i.i.d. property. Given the data stream $\{(x_t, y_t)\}_{t \in \mathbb{N}}$, we fit our model online and measure the mean squared error at the end of learning on a holdout test set, which contains independent samples across all $S_t$. As a comparison, we use neural networks to fit online as well as in batch and plot the errors under different correlation levels in Figure 4. The results show that our method outperforms neural networks in two aspects. First, when the data is i.i.d. ($d = 0$), Losse-FTL achieves a much lower error than NN-online. This indicates neural networks with gradient descent have low sample efficiency, even on stationary data. Training NNs using batch samples alleviates the issue and reaches a similar performance to ours. Second, our model consistently attains very low error across all correlation levels, showing its capacity of non-forgetting online learning. In contrast, neural networks learned with both batch and online updates incur increasingly high errors when $d$ is large, indicating catastrophic forgetting in the presence of data nonstationarity. More experimental details can be found in Appendix C.2.

## 5 EMPIRICAL RESULTS ON REINFORCEMENT LEARNING

In this section, we will demonstrate that world models built with Losse-FTL can be accurately learned online, outperforming several NN-based world model baselines and improving the data efficiency of RL agents. We first introduce the settings and baselines in Section 5.1, and then present our results in Section 5.2.

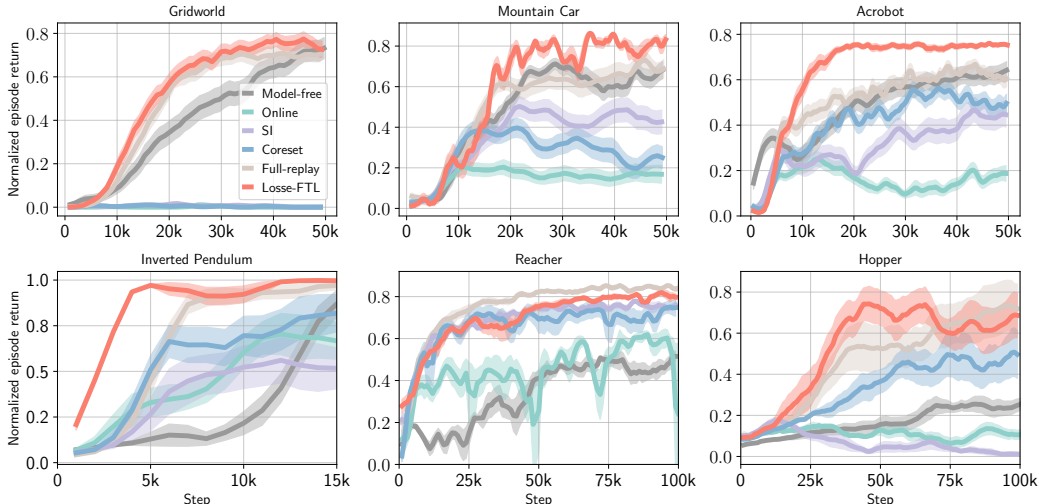

**Figure 5:** Learning curves showing normalized episode return. We compare our method with five baselines on (*top*) discrete control and (*bottom*) continuous control benchmarks. Solid curves depict the means of multiple runs with different random seeds, while shaded areas represent standard errors.

## 5.1 SETTINGS AND BASELINES

We employ the Dyna MBRL architecture and restrict our study to the model learning part, keeping the base agent's value (and policy) learning untouched. We compare Losse-FTL world models learned online with their NN-based counterparts, using the same model-free base agent. We introduce the baselines below. The algorithm and more experimental details can be found in Appendix D.

**Model-free**. The model-free base agent used in Dyna. We use two strong off-policy agents: DQN (Mnih et al., 2015) for discrete action space and SAC (Haarnoja et al., 2018) for continuous action space. All other settings below are MBRL coupled with either DQN or SAC.

**Full-replay**. We use neural networks to build the world model and train the NNs over all collected data till convergence. Before the training loop starts, the data is split into train and holdout sets with a ratio of 4:1. The loss on the holdout set is measured after each epoch. We stop training if the loss does not improve over 5 consecutive epochs. The NNs are trained continuously without resetting the weights (Nagabandi et al., 2020). This is the closest setting to FTL for NNs, but with growing computational cost as the data accumulates.

**Online**. We train NN-based world models in a stream learning fashion, where only a mini-batch of 256 transitions is kept for training. However, we still update the model for 250 gradient descent steps on each mini-batch, unlike our method which only needs a single pass of the interaction trajectory.

**SI**. This is similar to Online, but the NN is trained with Synaptic Intelligence (SI) (Zenke et al., 2017), a regularization-based continual learning (CL) method to overcome catastrophic forgetting. However, there is no explicit task boundary defined in the RL process, thus we adapt SI to treat each training sample as a new task.

**Coreset**. This refers to rehearsal-based CL methods (Lopez-Paz & Ranzato, 2017; Chaudhry et al., 2019) that keep an important subset of experiences for replay. There are different ways to decide whether to keep or remove a data point. We use Reservoir sampling (Vitter, 1985), which uniformly samples a subset of items from a stream of unknown length. Similar to Full-replay, data split and early stopping strategies are adopted for each training epoch.

## 5.2 RESULTS

The learning curves are presented in Figure 5, where we compare Losse-FTL with aforementioned baselines on 6 environments. We first revisit the discrete control Gridworld environment, which is used for illustration in Figure 1. This environment is a variant of the one introduced by Peng & Williams (1993) to test Dyna-style planning. The agent is required to navigate from the starting lo-

cation to the goal position without being blocked by the barrier. A small offset is added on each step to make the state space continuous. In such navigation tasks, intuitively, the world model needs to learn from extremely nonstationary data. When the agent evolves from randomly behaving to nearly optimal, its state visitation changes dramatically, making all NN-based world models completely fail to learn due to catastrophic forgetting, unless experiences are fully replayed (see the first plot in Figure 5). The poorly learned models are unable to provide useful planning, leading to almost zero performance. In contrast, a world model based on Losse-FTL can be learned accurately and improves the data efficiency over Model-free.

In the other two general discrete control tasks, Mountain Car and Acrobot, the data nonstationarity still exists but may be less severe than that in navigation tasks. The results show that Losse-FTL consistently outperforms all baselines and brings sample efficiency improvement.

For continuous control tasks from Gym Mujoco (Todorov et al., 2012), we can observe that Online NN fails to capture the dynamics for all environments, resulting in even worse performance than the Model-free baseline. While SI and Coreset can help reduce forgetting to some extent, their performances vary across tasks and are inferior to Full-replay in all cases. Losse-FTL, however, achieves on-par or even better performance than Full-replay, which is a strong baseline that continuously trains a deep neural network on all collected data until convergence to pursue FTL. These positive results verify the strength of Losse-FTL for building an online world model.

**Model update efficiency**. Among continual learning methods with neural networks, rehearsal-based ones are closer to the FTL objective and usually demonstrate strong performance (Chaudhry et al., 2019; Boschini et al., 2022). It seems that Coreset can offer a good balance between computation cost and non-forgetting performance with a properly set buffer size. We hence ablate Coreset with different replay sizes and compare their performance with Losse-FTL. Figure 6 shows both the sample efficiency and

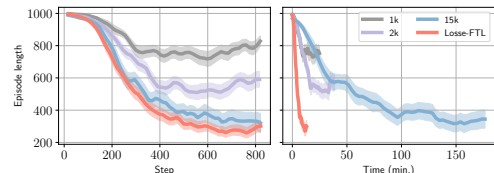

**Figure 6:** *(Left)* Sample and *(Right)* wall-clock efficiency comparison between world models learned with Losse-FTL and Coreset of different replay sizes.

wall-clock efficiency on the Gridworld environment. Coreset with a small replay size leads to poor asymptotic performance while being relatively more efficient. Increasing the replay size indeed contributes to better results at the cost of more computation. In comparison, Losse-FTL updates the model purely online without any replay buffer and achieves the best performance with the best wall-clock efficiency.

## 6    LIMITATION AND FUTURE WORK

Currently, we have not validated our method on problems with very high-dimensional state space such as Humanoid, or tasks with image observations. They are expected to be challenging for linear models to directly work on. Extending Losse-FTL to handle such large-scale problems, perhaps using pre-trained models to obtain compact sparse encoding, is a potential avenue for future work.

## 7    CONCLUSION

In this work, we investigate the problem of online world model learning using Dyna. We first demonstrate that NN-based world models suffer from catastrophic forgetting when used for MBRL because the data collected in the RL process is innately nonstationary. As a result, re-training over all previous data is adopted by common MBRL methods, which is of low efficiency. Through the lens of online learning, we uncover the implicit connection between NN re-training and FTL, which allows us to formulate online model learning with linear regressors to achieve incremental update. We devise locality sensitive sparse encoding (Losse), a high-dimensional non-linear random feature generator that can be coupled with a linear layer to produce models with greater capacities. We provide a sparsity guarantee for Losse, controlled by two parameters, $\rho$ and $\lambda$, which can be adjusted to develop an efficient sparse update algorithm. We test our method in both supervised learning and MBRL settings, with the presence of data nonstationarity. The positive results verify that Losse-FTL is capable of learning accurate world models online, which enhances both the sample and computation efficiency of reinforcement learning.

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

## A ON FEATURE SPARSITY

Feature sparsity has been pursued to reduce forgetting dating back to McCloskey & Cohen (1989); French (1991). The core idea is that catastrophic forgetting occurs due to the overlap of distributed representations in NNs, hence it can be reduced by reducing the overlap. Sparse features are local features with less overlap, so they should alleviate the forgetting issue. Recent works have demonstrated positive results by enforcing sparse hidden features in NNs either by regularization (Liu et al., 2019; Hernandez-Garcia & Sutton, 2019) or by deterministic activation functions (Pan et al., 2021; Lan & Mahmood, 2023). However, we emphasize that, in our work, *non-forgetting is ensured by FTL* and the high-dimensional sparse encoding is mainly for representational capacity and computational efficiency, though it also reduces feature interference as a byproduct.

We first conduct theoretical analysis similar to Lan & Mahmood (2023) to understand how sparsity contributes to non-forgetting, then present empirical results to show sparsity alone is not sufficient to guarantee non-forgetting.

Assume we are regressing a scalar target given an input vector using squared loss: $\ell(f, \mathbf{x}, y) = (f_{\mathbf{w}}(\mathbf{x}) - y)^2$, where $f$ is our predictor with parameters $\mathbf{w}$. When a new data sample $\{\mathbf{x}_t, y_t\}$ arrives, we want to adjust the parameters $\mathbf{w}$ to minimize $\ell(f, \mathbf{x}_t, y_t)$ with gradient descent:

$$\mathbf{w}' - \mathbf{w} = \Delta \mathbf{w} = -\alpha \nabla_{\mathbf{w}} \ell(f, \mathbf{x}_t, y_t) \tag{8}$$

The change of the parameters will lead to the change of the predictor's output, and by the first-order Taylor expansion:

$$\begin{aligned} f_{\mathbf{w}'}(\mathbf{x}) - f_{\mathbf{w}}(\mathbf{x}) &\approx \nabla_{\mathbf{w}} f_{\mathbf{w}}(\mathbf{x}) (\mathbf{w}' - \mathbf{w}) \\ &= -\alpha \nabla_{\mathbf{w}} f_{\mathbf{w}}(\mathbf{x}) \nabla_{\mathbf{w}} \ell(f, \mathbf{x}_t, y_t) \\ &= -\alpha \nabla_f \ell(f, \mathbf{x}_t, y_t) \langle \nabla_{\mathbf{w}} f_{\mathbf{w}}(\mathbf{x}), \nabla_{\mathbf{w}} f_{\mathbf{w}}(\mathbf{x}_t) \rangle \end{aligned} \tag{9}$$

We assume non-zero loss so $\nabla_f \ell(f, \mathbf{x}_t, \mathbf{y}_t) \neq 0$ and analyze $\langle \nabla_{\mathbf{w}} f_{\mathbf{w}}(\mathbf{x}), \nabla_{\mathbf{w}} f_{\mathbf{w}}(\mathbf{x}_t) \rangle$. For linear models as in our case, $f_{\mathbf{w}}(\mathbf{x}) = \mathbf{w}^\top \phi(\mathbf{x})$, thus $\langle \nabla_{\mathbf{w}} f_{\mathbf{w}}(\mathbf{x}), \nabla_{\mathbf{w}} f_{\mathbf{w}}(\mathbf{x}_t) \rangle = \langle \phi(\mathbf{x}), \phi(\mathbf{x}_t) \rangle$. Then we could investigate the following question: after we adjust the parameters (Equation 8) based on the new sample, what's the output difference (Equation 9) at any $\mathbf{x}$ for $\mathbf{x}$ that is *similar* or *dissimilar* to $\mathbf{x}_t$?

1. For $\mathbf{x}$ similar to $\mathbf{x}_t$, we expect $\langle \phi(\mathbf{x}), \phi(\mathbf{x}_t) \rangle \neq 0$ to yield improvement and generalization. This will hold since a proper feature encoder $\phi(\cdot)$ should approximately preserve the similarity in the original space.

2. For $\mathbf{x}$ dissimilar to $\mathbf{x}_t$, we hope $\langle \phi(\mathbf{x}), \phi(\mathbf{x}_t) \rangle \approx 0$ to reduce interference (therefore less forgetting). The claim is that if $\phi(\cdot)$ produces sparse features, it is highly likely that the inner product is approximately zero, hence meeting our expectation.

However, we argue that feature sparsity is not sufficient to mitigate forgetting, because the probability that there is no single overlap between representations is low even if the feature is sparse. In Figure 7, we show the mean squared errors for the piecewise random walk stream learning task introduced in Section 4.2. We use our Losse representation with different sparsity levels as features and compare our sparse FTL update with gradient descent update. We can observe that the gradient descent update relying on feature sparsity still suffers from catastrophic forgetting until $\lambda = 30$, which corresponds to a 9000-dimensional feature with 40 nonzero entries — an extremely high sparsity level. Nevertheless, increasing sparsity indeed helps sparse features to reduce interference when using gradient descent (comparing the absolute values of MSEs across different $\lambda$). In contrast, *Losse-FTL ensures non-forgetting incrementally by finding the optimal fitting for all experienced samples, even with the presence of feature overlap*. Notably, when Losse gets more sparse, we can observe the error for Losse-FTL decreases, verifying our design that the high dimensional sparse feature brings more fitting capacity while maintaining the same computational cost with our sparse update.

## B ANALYTIC CONTINUAL LEARNING

Our work is closely related to a paradigm called *analytic continual learning*, where the closed-form solution, if obtainable, is employed in training a continual learner. A recent line of research explores

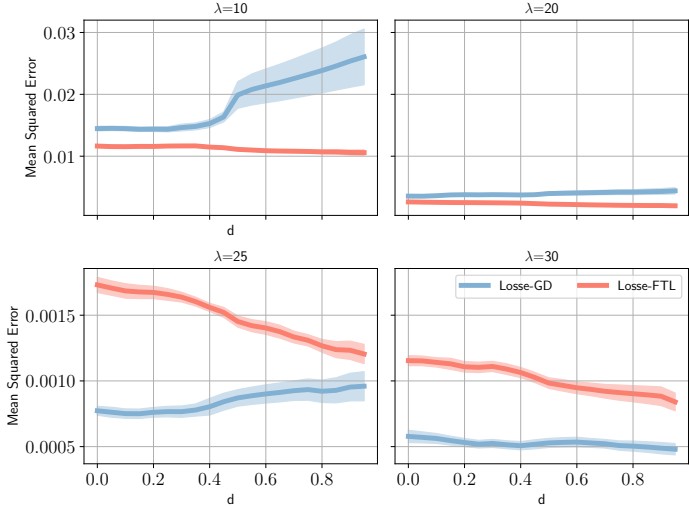

**Figure 7:** Mean squared errors on piecewise random walk stream learning. The results compare the FTL update to the gradient descent update using Losse features with different sparsity levels. Note that gradient descent (GD) update uses a mini-batch of 50 samples for each update, resulting in lower errors than FTL's when the feature dimension gets higher.

this topic focusing on class-incremental learning (Zhuang et al., 2022; 2023; 2024). In particular, Zhuang et al. (2022) employ a pre-trained neural network to extract feature vectors $\boldsymbol{X}_k^{(\mathrm{fe})}$ after a process called feature expansion, and then learn the weights of a fully connected layer $\hat{\boldsymbol{W}}_{\mathrm{FCN}}$ on the frozen features for class-incremental classification. The weights at round $k$ can be solved analytically in a recursive manner utilizing the Woodbury matrix identity:

$$\hat{\boldsymbol{W}}_{\mathrm{FCN}}^{(k)} = \left[ \ \hat{\boldsymbol{W}}_{\mathrm{FCN}}^{(k-1)} - \boldsymbol{R}_k \boldsymbol{X}_k^{(\mathrm{fe})\top} \boldsymbol{X}_k^{(\mathrm{fe})} \hat{\boldsymbol{W}}_{\mathrm{FCN}}^{(k-1)} \boldsymbol{R}_k \boldsymbol{X}_k^{(\mathrm{fe})\top} \boldsymbol{Y}_k^{\mathrm{train}} \ \right], \tag{10}$$

where $\boldsymbol{Y}_k^{\mathrm{train}}$ is the label for class $k$, and $\boldsymbol{R}_k = \left( \sum_{i=0}^{k} \boldsymbol{X}_i^{(\mathrm{fe})\top} \boldsymbol{X}_i^{(\mathrm{fe})} + \gamma \boldsymbol{I} \right)^{-1}$ is the regularized feature auto-correlation matrix (similar to the inverse of $\boldsymbol{A}_t = \boldsymbol{\Phi}_t^\top \boldsymbol{\Phi}_t$ in our formulation), which also has a recursive form:

$$\boldsymbol{R}_k = \boldsymbol{R}_{k-1} - \boldsymbol{R}_{k-1} \boldsymbol{X}_k^{(\mathrm{fe})\top} \left( \boldsymbol{I} + \underbrace{\boldsymbol{X}_k^{(\mathrm{fe})} \boldsymbol{R}_{k-1} \boldsymbol{X}_k^{(\mathrm{fe})\top}}_{N_k \times N_k} \right)^{-1} \boldsymbol{X}_k^{(\mathrm{fe})} \boldsymbol{R}_{k-1}. \tag{11}$$

Though Zhuang et al. (2022) have achieved state-of-the-art class-incremental learning performance, there are a few limitations. First, a deep network must be pre-trained and frozen to serve as an expressive feature encoder, which is simply infeasible in online settings. Second, Equation 11 involves inverting a matrix whose size grows with the number of samples, limiting its scalability. Third, the learning system requires the knowledge of task boundaries for matrix decomposition, which may not be accessible in many real-world scenarios. In contrast, our work develops an expressive feature encoder that does not rely on pre-training, proposes a constant-time update rule exploiting the feature sparsity, and applies to reinforcement learning where the nonstationarity appears without explicit task boundaries.

## C SUPERVISED LEARNING DETAILS

In this section, we provide more details regarding the supervised learning experiments on image denoising (Appendix C.1) and piecewise random walk (Appendix C.2).

### C.1 IMAGE DENOISING

We add pixel-wise Gaussian noise with $\mu = 0, \sigma = 0.3$ to normalized images from the MNIST dataset (Deng, 2012), and create train and test splits with ratio $9 : 1$. To test the feature representation

on different difficulties, we crop patches from the center with different sizes, ranging from $2 \times 2$ to $7 \times 7$. Figure 8 shows some sample pairs. For different feature encoding methods compared

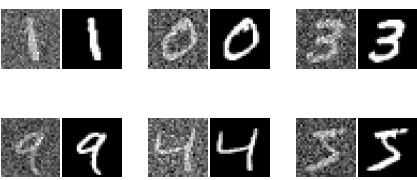

**Figure 8:** Sample pairs for the image denoising task.

in Section 4.1, we limit the number of non-zero feature entries up to 80 for fair comparison. This means 2-d binning with 20 grids for Losse-FTL. The number of bins affects the performance slightly, due to different granularity for generalization. Hence we sweep it over $\{5, 6, 7, 8, 9\}$. We observe that the scale of the standard deviation of the Gaussian random projection matrix affects the final performance significantly for random ReLU and random Frouier features, so we select the best one from a sweep over $\{0.1, 0.3, 0.5, 1, 5, 10\}$. The linear layer is optimized using Adam (Kingma & Ba, 2015) with a learning rate 0.0001. We repeated experiments for all settings for 5 times and report the mean scores.

## C.2 PIECEWISE RANDOM WALK

Recall that observations in the process are sampled from a Gaussian $X_t \sim \mathcal{N}(S_t, \beta^2)$ with $\{S_t\}_{t \in \mathbb{N}}$ being a Gaussian first order auto-regressive random walk $S_{t+1} = (1 - c)S_t + Z_t$, where $c \in (0, 1]$ and $Z_t \sim \mathcal{N}(0, \sigma^2)$. The equilibrium distribution of $\{X_t\}_{t \in \mathbb{N}}$ is also a Gaussian with $\mathbb{E}[X_t] = 0$ and variance $\xi^2 = \beta^2 + \frac{\sigma^2}{2c - c^2}$. We can use a single parameter $d \in [0, 1)$, which we call correlation level, to control the variance:

$$c = 1 - \sqrt{1 - d},$$
$$\sigma^2 = d^2 \left(\frac{B}{2}\right)^2,$$
$$\beta^2 = (1 - d)\left(\frac{B}{2}\right)^2,$$

where $B$ gives a high probability bound. See Pan et al. (2021) for detailed derivation.

We visualize sampling trajectories of $X_t$ for $B = 1$ $d \in [0, 1)$ in Figure 9.

For neural networks trained in Section 4.2, we use 2-layer MLP with 50 hidden units. For NN-Batch, we take 50 samples from each $X_t$ and use mini-batch gradient descent to update the weights. Adam optimizer is used with the best learning rate swept from $\{5 \times 10^{-6}, 1 \times 10^{-5}, 5 \times 10^{-5}, \ldots, 1 \times 10^{-2}\}$. For Losse-FTL, we use 2-d binning with 10 bins for each grid and use 10 grids to construct the final feature. We run all experiments for 50 independent seeds and report the means and standard errors.

## D REINFORCEMENT LEARNING DETAILS

**Learning algorithm**. We present the detailed Dyna-style algorithm based on Losse-FTL in Algorithm 3, where the world models are learned online. For NN-based settings, line 4 of Algorithm 3 is replaced by training a neural network (optionally with replay buffer or other continual learning techniques).

**Settings for discrete control tasks**. We use $\kappa = 30$, $\rho = 2$, and $\lambda = 10$ for Losse-FTL, and 3-layer MLPs with 32 hidden units for neural networks. For Coreset we maintain a buffer with size 100, and for SI we use the best $\xi$ and $c$ from a grid search. The DQN parameters are updated using real data with an interval of 4 interactions. In MBRL, 16 planning steps are conducted after each real data update. Both real data updates and planning updates use a mini batch of 32. The learning rate of DQN is swept from $\{1 \times 10^{-2}, 3 \times 10^{-3}, 1 \times 10^{-3}, \ldots, 1 \times 10^{-5}\}$ for all configurations. All world

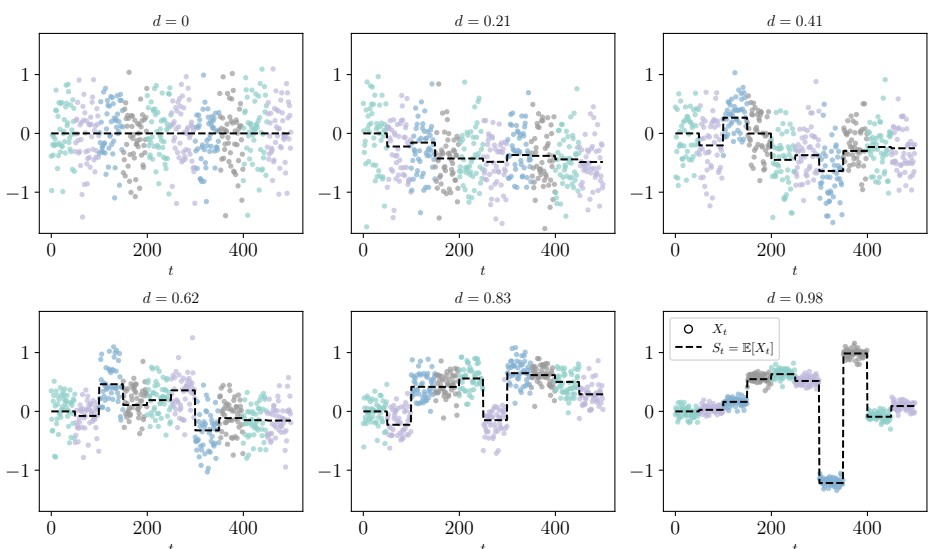

**Figure 9:** Visualization of piecewise random walk sampling trajectories with different correlation levels $d$.

---

**Algorithm 3** Dyna MBRL with Losse-FTL

---

**Require:** environment $\mathcal{M}$, agent $\pi$, world model $\hat{f}$, state search control $c$, model unroll length $k$, model experiences $\mathcal{D}_m$
1: **for** epoch $\in \{1, \ldots, E\}$ **do**
2:     **for** interaction $\in \{1, \ldots, K\}$ **do**
3:         $\mathbf{s}, \mathbf{a}, \mathbf{s}', r \leftarrow \texttt{rollout}(\mathcal{M}, \pi)$
4:         Update $\hat{f}$ with $(\mathbf{s}, \mathbf{a}, \mathbf{s}', r)$ (Algorithm 2)              ▷ incremental model updates
5:     **end for**
6:     **for** planning $\in \{1, \ldots, N\}$ **do**
7:         $\tilde{\mathbf{s}} \leftarrow c$                                            ▷ uniform state sampling
8:         $\tilde{\mathbf{a}}, \hat{\mathbf{s}}', \hat{r} \leftarrow \texttt{ModelUnroll}(\hat{f}, \tilde{\mathbf{s}}, \pi, k)$       ▷ short-horizon on-policy unroll
9:         $\mathcal{D}_m \leftarrow (\tilde{\mathbf{s}}, \tilde{\mathbf{a}}, \hat{\mathbf{s}}', \hat{r})$
10:    **end for**
11:    **for** learning $\in \{1, \ldots, G\}$ **do**
12:        Update DQN or SAC on a batch of model data $\mathcal{B} \sim \mathcal{D}_m$
13:    **end for**
14: **end for**

---

models are updated every 25 environment steps to accelerate experiments. We run all experiments for 30 independent seeds and report the means and standard errors.

**Settings for continuous control tasks.** We use $\kappa = 300, \rho = 2$, and $\lambda = 10$ for Losse-FTL, and 4-layer MLPs with 400 hidden units for neural networks following Janner et al. (2019). Our implementation follows closely the official codebase[2] of MBPO (Janner et al., 2019), and also adopts the same hyper-parameters for both MBPO and SAC. The replay size of Coreset is 5000, and the hyper-parameters for SI are chosen from a grid search. All world models are updated every 250 environment steps to accelerate experiments. We run all experiments for 5 independent seeds and report the means and standard errors.

## E IMPLEMENTATION DETAILS

We provide more implementation details of Losse-FTL in this section. Referring to Figure 3, the input $\mathbf{x}_t$ is first pre-processed to be bounded between $[-3, 3]$, and then projected with a matrix

---

[2]https://github.com/jannerm/mbpo

containing values sampled from an isotropic Gaussian with $\mu = \mathbf{0}$ and $\mathbf{\Sigma} = \frac{1}{c}\mathbf{I}$, where $c$ is the fan-in dimension, i.e., $S + A$. This ensures the projected values $\sigma(\mathbf{x}_t)$ are bounded with high probability and facilitates a more even bin utilization. In computing the sparse update at line 7 of Algorithm 2, we use $(\mathbf{A}_{t,ss} + \varepsilon\mathbf{I})^{-1}$ for some small $\varepsilon > 0$ to ensure its invertibility. We provide Jax-based Python codes in Listing 1.

```python
import math
from typing import NamedTuple, Tuple

import chex
import jax
import jax.numpy as jnp
import numpy as np

class LosseParams(NamedTuple):
    count: jax.Array
    projection: jax.Array
    xtx: jax.Array
    xty: jax.Array
    w: jax.Array

def _to_1d_index(indices, offsets, n_feat, bin_dim, n_bins):
    """Compute the flattened index into the weight matrix."""
    n_grids_per_lsh = (n_bins + 1) ** bin_dim
    indices = jnp.reshape(indices, (-1, bin_dim, n_feat))
    offsets = jnp.reshape(offsets, (-1, bin_dim, n_feat))
    indices = jnp.stack([indices, indices + 1], axis=-1)  # [-1, bin_dim, n_feat, 2]
    values = jnp.stack([1.0 - offsets, offsets], axis=-1)  # [-1, bin_dim, n_feat, 2]
    multiplier = jnp.power(n_bins + 1, jnp.arange(bin_dim - 1, -1, -1))
    indices *= multiplier[:, None, None]
    # shape = (-1, n_feat, ) + (2,) * bin_dim
    shape_suffix = [tuple(*p) for p in np.split(np.eye(bin_dim, dtype=np.int32) + 1,
    ↪  bin_dim)]
    indices = sum(jnp.reshape(indices[:, i], (-1, n_feat, *suffix)) for i, suffix in
    ↪  enumerate(shape_suffix))
    values = math.prod(jnp.reshape(values[:, i], (-1, n_feat, *suffix)) for i, suffix in
    ↪  enumerate(shape_suffix))
    # both indices and values has the shape (-1, n_feat, *(2,)*bin_dim) now.
    indices += jnp.expand_dims(
        n_grids_per_lsh * jnp.arange(n_feat), axis=tuple(range(-bin_dim, 1, 1))
    )  # expand 1 dim in the front and bin_dim in the back.
    indices = jnp.reshape(indices, (-1, n_feat * 2**bin_dim))
    values = jnp.reshape(values, (-1, n_feat * 2**bin_dim))
    return indices, values

class Losse:
    """Linear regressor with LOcality Sensitive Sparse Encoding (Losse).

    We update the linear weights online sparsely following Algorithm.2 in the paper, i.e.,
    ↪  computing the incremental closed-form solution based on newly incoming data points.
    """

    def __init__(
        self,
        inout_dims: Tuple[int, int],
        num_features: int,
        num_bins: int,
        bin_dim: int,
        eps: float,
    ) -> None:
        self.num_features = num_features
        self.num_bins = num_bins
        self.bin_dim = bin_dim
        self.inout_dims = inout_dims
        self.eps = eps
        n_edges = num_bins + 1
        n_grids_per_lsh = (n_edges + 1) ** bin_dim
        self.d = n_grids_per_lsh * num_features

    def init(self, rng: jax.random.PRNGKey) -> LosseParams:
        input_dim = self.inout_dims[0]
        output_dim = self.inout_dims[1]
        std = 1 / jnp.sqrt(input_dim)
        projection = std * jax.random.truncated_normal(
            rng,
            -2,
```

```python
 70                2,
 71                (input_dim, self.num_features * self.bin_dim),
 72            )
 73            return LosseParams(
 74                count=jnp.array(0, dtype=jnp.int64),
 75                projection=projection,
 76                xtx=jnp.zeros((self.d * self.d,), dtype=projection.dtype),
 77                xty=jnp.zeros((self.d, output_dim), dtype=projection.dtype),
 78                w=jnp.zeros((self.d, output_dim), dtype=projection.dtype),
 79            )
 80
 81        def update(
 82            self,
 83            params: LosseParams,
 84            x: jax.Array,
 85            y: jax.Array,
 86        ) -> LosseParams:
 87            chex.assert_tree_shape_prefix((x, y), (1,))  # assert non-batched
 88            indices, values = self._indices_and_values(params.projection, x)
 89            params = self._update_memory(params, indices, values, y)
 90            params = self._update_w(params, indices)
 91            return params
 92
 93        def predict(self, params: LosseParams, x: jax.Array):
 94            indices, values = self._indices_and_values(params.projection, x)
 95            output = params.w[indices] * values[..., None]
 96            return output.sum(1)
 97
 98        def _indices_and_values(
 99            self,
100            projection: jax.Array,
101            x: jax.Array,
102        ):
103            h = jnp.matmul(x, projection)
104            h = jax.nn.sigmoid(h)
105            h = jnp.clip(h, 0, 1) * self.num_bins
106            indices = jnp.floor(h).astype(jnp.int32)
107            offsets = h - indices
108            indices, values = _to_1d_index(
109                indices,
110                offsets,
111                self.num_features,
112                self.bin_dim,
113                self.num_bins,
114            )
115            return indices, values
116
117        def _update_memory(
118            self,
119            params: LosseParams,
120            indices: jax.Array,
121            values: jax.Array,
122            y: jax.Array,
123        ) -> LosseParams:
124            chex.assert_equal_shape_prefix((indices, values, y), prefix_len=1)
125            xtx_indices = (indices * self.d)[:, :, None] + indices[:, None, :]
126            xtx_indices = xtx_indices.flatten()
127            xty_indices = indices.flatten()
128            xtx_updates = values[:, :, None] * values[:, None]
129            xtx_updates = xtx_updates.flatten()
130            xty_updates = values[:, :, None] * y[:, None, :]
131            xty_updates = xty_updates.reshape(-1, y.shape[-1])
132            return params._replace(
133                xtx=params.xtx.at[xtx_indices].add(xtx_updates),
134                xty=params.xty.at[xty_indices].add(xty_updates),
135                count=params.count + y.shape[0],
136            )
137
138        def _update_w(
139            self,
140            params: LosseParams,
141            indices: jax.Array,
142        ) -> LosseParams:
143            indices = indices.flatten()
144            sub_indices = (indices * self.d)[:, None] + indices[None, :]
145            sub_xtx = jnp.reshape(params.xtx[sub_indices], [indices.shape[0]] * 2)
146            sub_xty = params.xty[indices]
147            a = sub_xtx
148            sub_xtxw = jnp.matmul(
149                jnp.reshape(params.xtx, (self.d, self.d))[indices],
150                params.w,
```

```
151                )
152                b = sub_xty - sub_xtxw + jnp.matmul(sub_xtx, params.w[indices])
153                a_norm = a / params.count + self.eps * jnp.eye(len(a))
154                b_norm = b / params.count
155                solution = jnp.linalg.solve(a_norm, b_norm)
156                return params._replace(w=params.w.at[indices].set(solution))
157
158
159     losse = Losse(
160         inout_dims=(1, 1),
161         num_features=50,
162         num_bins=10,
163         bin_dim=2,
164         eps=1e-5,
165     )
166
167     losse.init = jax.jit(losse.init)
168     losse.update = jax.jit(losse.update, donate_argnums=(0,))   # donate to avoid copy
169     losse.predict = jax.jit(losse.predict)
```

**Listing 1:** Jax-based Python codes for learning an online linear regressor with Losse-FTL.

