# OpenReview forum: "Locality Sensitive Sparse Encoding for Learning World Models Online"
_ICLR.cc/2024/Conference — ICLR 2024 poster_

### Official Review · Reviewer_CrA3 · 2023-10-24

**Soundness:** 2 fair
**Presentation:** 2 fair
**Contribution:** 3 good
**Rating:** 6
**Confidence:** 4

**Summary:**

This paper presents a world model for model-based reinforcement learning (RL) which can be learned online and does not
require full retraining on all previous data.
The authors highlight that training world models is subject to issues arising in continual learning.
That is, each sequential data set can be interpreted as coming from a new task and thus the world model needs to be retrained
after each agent-environment interaction.
This is because the data collected via agent-environment interaction is non-stationary.
They propose a world model based on a linear regression model which uses high-dimensional nonlinear features.
Importantly, the linear model can be updated given new data whilst retaining good predictive performance on old data.
They compare their feature encoding method to other feature encoding techniques in an image denoising task on MNIST.
They then evaluate their method's ability to handle training data covariate shift in an artificial online learning experiment.
Finally, they evaluate their method's ability to combat non-stationary data in model-based RL.

**Strengths:**

This paper addresses an important problem in model-based RL, which is likely a problem that must be solved for developing lifelong agents.
The method for updating the linear model online is simple but appears to be effective.
This is also the first time I have seen a non-trainable encoder used for world models.
It is very common to see world models with NN encoders and NN transition models operating in the encoder's latent space.
Typically the dynamic model operates on a latent state which is lower dimensional than the high-dimensional observations.
Perhaps I am not aware of the relevant literature, but this seems like an interesting and original idea.

**Weaknesses:**

This paper has two main weaknesses.
Firstly, there is no comparison to state-of-the-art MBRL strategies that use world models, e.g. Dreamer/TD-MPC.
As such, there is no experiment highlighting the main issue that the paper is trying to address:
that NN-based world models suffer from catastrophic interference due to non-stationary data.
Second, all of the RL experiments are in simple RL environments.
From the current results, it is impossible to know how practically useful this world model is.
There is no discussion about its limitations nor is there a comparison to other model-based RL algorithms that use world models.
Sure the proposed method works on some simple RL environments but can it scale to difficult environments like humanoid and can it handle image-based observations?
It is OK if the method cannot do this but it should be addressed in the text.
Moreover, it should be made clear what benefit it does have over other world model methods (like Dreamer).
For example, I'd like to see a state-of-the-art world model method (like Dreamer) performing poorly/failing because it cannot handle non-stationary data.

I also have questions regarding the training of the NNs in the experiments.
Did the full-replay experiment involve resetting the neural network's weights? If so, what initialization was used?
When was the NN training stopped? Was the data split into train/validation sets and used to stop training when the
validation loss stopped improving?
The paper needs more details to explain exactly how this was implemented.
In my experience, these steps are important to ensure the NNs don't overfit on early data sets.

I am also unsure why the full-replay strategy (which is model-based), does not appear to have better sample
efficiency than the model-free experiment. Am I missing something here?
Perhaps this is an interesting point for discussion. Do the high-dimensional features sacrifice sample efficiency
in favour of formulating a linear model which can handle the non-stationary data?
I'm not sure if this is correct.
My main point here is that the paper has not answered all of my questions about the method.

The experiments tell the first part of a nice story.
Table 1 compares to other encoding methods and Fig. 3 clearly shows how the method handles covariate shift better than NNs.
Fig. 4 also acts as a nice ablation for comparing the method to other CL strategies within the same set-up.
However, the experiments section lacks a comparison to other MBRL strategies which use world models.
In particular, there is no experiment highlighting the main issue the paper is trying to address.
That is, there is no MBRL experiment failing due to the non-stationarity of the training data.

Minor comments and corrections:
- The abstract is very long. I would recommend shortening it.
- Sections shouldn't lead straight into subsections (Section 4/4.1, 5/5.1, B/B.1, C/C.1). There should be text explaining what the reader can expect to read in the section.
- In the first paragraph of Section 2.2, the reward function is defined as $R(\mathbf{s}, \mathbf{a}, \mathbf{s}')$ but then in the optimal policy equation you use $R(\mathbf{s}, \mathbf{a})$.
- Fourth line of Section 2.2 the initial state distribution is $\rho$ but earlier it is $\rho_0$.
- Third paragraph of Section 2.2 - "We firstly formulate" should be "We first formulate".
- Section 2.1 - "When the input is a convex set $\mathcal{S}$, the prediction a vector $\mathbf{w}_{t} \in \mathcal{S}$". This sentence doesn't read properly.
- $\rho$ is used to denote the initial state distribution and to denote the dimension of the grids in Section 3.2.
- What is the value of $\delta$ in Fig. 1?
- What are $\pi_0$, $\pi_t$ and $\pi_{t'}$?
- The first sentence of the abstract says model-based RL has better sample efficiency. Better than what? It's model-free counterparts?
- It is unusual to end the paper with a section titled "Summary". I recommend changing this to "Conclusion".

**Questions:**

- What are the limitations of the proposed method? Can it handle image observations? Can it scale to difficult environments like Humanoid?
- Why haven't you compared to any other world model algorithms? E.g. Dreamer, TD-MPC.
- How was the NN full-replay experiment implemented? Did the full-replay experiment involve resetting the neural network's weights? If so, what initialization was used? When was the NN training stopped? Was the data split into train/validation sets and used to stop training when the validation loss stopped improving?

---

> ### Author Response · Authors · 2023-11-15
> **Response to Reviewer CrA3 (1/2)**
>
> Thank you very much for appreciating the originality of our paper, as well as your detailed feedback and suggestions. We have updated the paper with text changes highlighted in yellow, and elaborate on each of the questions below.
>
> **1. There is no experiment highlighting the main issue that the paper is trying to address: NN-based world models suffer from catastrophic interference due to non-stationary data.**
>
> Experiments in Section 5 are designed to compare our method with different variants of NN-based world models while keeping the value/policy learning the same. Except for the “Model-free” setting, all other baselines train NN-based world models. From Figure 4, we can observe that NN-based world models trained online (referred to as “Online” in the legend) show inferior performance. This is an indication that they suffer from catastrophic interference due to non-stationary data. As a result, the online learned NN world model could only generate synthetic transitions of poor quality, deteriorating the agent performance. Applying continual learning techniques (such as “SI” and “Coreset”) alleviates the forgetting issue to some extent. And learning world models with our method outperforms other baselines for most environments.
>
> **2. Why aren't SoTA MBRL algorithms (e.g., Dreamer/TD-MPC) included for comparison?**
>
> First, we hope that we can get aligned to the point that NN-based MBRL suffers from catastrophic forgetting, including the SoTA algorithms. The arguments are following,
> 1. Catastrophic forgetting is a universal phenomenon for neural networks.
> 2. Existing end-to-end NN-based MBRL methods, such as Dreamer, TD-MPC, and MuZero, etc. rely heavily on techniques to make data more stationary, for example, maintaining a large replay buffer, updating the target network once in a while, or simultaneously running multiple environments. When these components are removed, they would fail to work.
> 3. However, when they fail we don't know how much is due to the forgetting in policy/value/model because all components are entangled and trained end-to-end in these methods.
>
> If we're aligned on the point that NN-based MBRL suffers from catastrophic forgetting due to non-stationary data, then what we're studying is that we focus on making the world model fully online, which relies on *none* of the above techniques. With this scope, we only compare our world models and NN-based world models using the Dyna architecture, because it decouples policy/value learning from model learning, making an apple-to-apple comparison on model learning possible.
>
> **3. Can your method work for more difficult tasks like Humanoid or handle image-based observations?**
>
> Currently, we have not validated it for more challenging tasks like you mentioned. In Humanoid, the dimension of the state is very high, posing a curse of dimensionality to our method. For image-based observations, finding a compact representation with a non-trainable encoder is non-trivial, and our attempts so far do not yield positive results. We have added a Limitation section on page 9 to address them in the paper.

---

> ### Author Response · Authors · 2023-11-15
> **Response to Reviewer CrA3 (2/2)**
>
> **4. NN training details for the Full-replay experiment.**
>
> The Full-replay experiments did not involve resetting the NN’s weights. We have tried resetting the weights but empirically found this required much more compute budget since it trained from scratch every time, which is impractical for a lifelong RL agent. Besides, it raises another design choice about when to reset, which may be not trivial since there are no explicit task boundaries.
>
> Instead, we “finetune” a single model continuously, with the train/holdout split (with a ratio of 8:2) to determine when to stop the training and avoid overfitting. We agree that training without resetting weights is likely to overfit early data sets, but empirically this seems okay. We closely follow the codes of MBPO (https://github.com/jannerm/mbpo), which also store all previous experiences and do periodic training without weights resetting, for our implementation. We have updated the paper and provided more details about NN training in Section 5.1 on page 8.
>
> **5. Why isn’t the Full-replay MBRL better than the model-free baseline?**
>
> The expectation that Full-replay MBRL should outperform the Model-free baseline is based on an assumption: an accurate world model is learned such that it can synthesize transitions on which planning updates accelerate value/policy learning. However, achieving so is non-trivial. As you have mentioned, successful training requires stopping training properly. Though we have employed a train/holdout split to determine when to stop, our parameters may not be optimally set. Apart from this, there is more brittleness in NN training in this case. For example, as you mentioned, not resetting the weights may make NNs overfit early data; but when to reset is hard to know because there are no explicit task boundaries; even if we know when, resetting and training from scratch is even more time-consuming and impractical for lifelong agents. In our experiments, we choose not to reset the weights for efficiency consideration, and this might be a source of training difficulty that leads to inferior Full-replay results, especially for Acrobot environment.
>
> **6. Other minor corrections.**
>
> Thank you very much for the detailed feedback! We have updated the paper to address all the mentioned issues.
>
> ---
> Thank you again for providing insightful comments which helped us to improve our paper, and we hope our responses and updates in the text were able to address any remaining concerns. Please do let us know if you have any further questions as well as what would be expected for score improvement.

---

> > ### Comment · Reviewer_CrA3 · 2023-11-18
> >
> > Thank you for the detailed comments and for updating your manuscript. I will increase my score to a 6.
> >
> > In my opinion, this paper doesn't isolate the issue of nonstationary data for world model training. I appreciate that model-based RL algorithms have a lot of moving parts, which can make it hard to isolate specific problems. I do think that training until the validation loss stops decreasing will 100% lead to the NN overfitting early in training. As such, the limited performance of Full-replay is likely due to primacy bias and not the nonstationary data. A simple solution is to add one extra baseline. A modification of the Full replay experiment where the NN is reset at each episode. Whilst I know this is not practical for a lifelong agent, it should be feasible for the simple environments reported in the paper. This would remove issues with primacy bias and allow the baseline to act as a sort of upper bound on model-based performance.
> >
> > When talking about world models, readers will immediately think of methods like Dreamer. My questions about Dreamer/TD-MPC arose because you refer to your method as a world model. I'd suggest the authors move away from the term world model and instead stick to Dyna. Also, consider adding this part of your response to the manuscript:
> >
> > "Existing end-to-end NN-based MBRL methods, such as Dreamer, TD-MPC, and MuZero, etc. rely heavily on techniques to make data more stationary, for example, maintaining a large replay buffer, updating the target network once in a while, or simultaneously running multiple environments. When these components are removed, they would fail to work.
> > However, when they fail we don't know how much is due to the forgetting in policy/value/model because all components are entangled and trained end-to-end in these methods."
> >
> > Given that you have now clearly stated your method's limitations, with regard to image-observations etc, I am OK without a direct comparison to Dreamer/TD-MPC.  I agree that comparing to dyna style methods makes a comparison of model learning easier.

---

> > > ### Author Response · Authors · 2023-11-19
> > >
> > > Thank you very much for the score improvement and for providing valuable feedback.
> > >
> > > We have updated the manuscript to include the quoted discussion about other NN-based MBRL methods (in Section 2.2), as well as putting the term “world model” in the context of Dyna wherever needed. We will add another baseline for Full-replay with weights reset to demonstrate the upper bound performance for NN models.

---

### Official Review · Reviewer_LJy1 · 2023-10-30

**Soundness:** 3 good
**Presentation:** 3 good
**Contribution:** 3 good
**Rating:** 8
**Confidence:** 3

**Summary:**

The paper presents a method for the online learning of a world model for model-based reinforcement learning (MBRL). To obtain efficient updates to the world model, the world model is expressed as a linear combination of a set of spare features. This efficiency allows online learning at a constant computational cost.

**Strengths:**

The work is well motivated in the introduction and a sufficient and clear background is provided for non-expert readers in the preliminaries section. The algorithms, definitions, etc. are mathematically rigorously presented.

A comprehensive set of experiments has been conducted demonstrating the efficacy of Losse-FTL

**Weaknesses:**

Significant discussion around catastrophic forgetting was mentioned in the introduction but little discussion is presented in the main text and left in the appendix.

“Example 3.1” could be a regular paragraph. Formatting this as an Example does not improve readability and is, in fact, the only Example in the entire paper.

**Questions:**

In figure (1): d(s_(t+1), f(st, at)) and \delta were not defined in the caption or anywhere obvious in the main text.

In eq (3), does || . ||^2_F denote the Frobenius matrix norm? It is only stated so after eq (5). It helps to have the notation introduced earlier here. Especially since “F” is the dimension of the feature space

---

> ### Author Response · Authors · 2023-11-15
> **Response to Reviewer LJy1**
>
> Thank you very much for highlighting our comprehensive experimental results, as well as your valuable feedback on the paper presentation. We have updated the paper with text changes highlighted in yellow, and elaborate on each of the questions below.
>
> **1. Significant discussion on catastrophic forgetting in the introduction but little discussion in the main text and left in the appendix.**
>
> Thanks for this suggestion to improve our paper presentation. We have updated the paper such that catastrophic forgetting is emphasized more in the main text, especially at the end of Section 3 on page 6. Due to the space constraints, however, we still use Appendix A for more detailed discussions.
>
> **2. Other issues regarding the notations.**
>
> We have updated accordingly in the paper.
>
> ---
> Thank you again for your feedback to help us refine the paper quality, and we hope our responses and updates in the text were able to address any remaining concerns.

---

### Official Review · Reviewer_2Y3F · 2023-10-31

**Soundness:** 2 fair
**Presentation:** 3 good
**Contribution:** 2 fair
**Rating:** 6
**Confidence:** 3

**Summary:**

The paper introduces a model-based reinforcement learning approach that utilizes a sparse representation-to-representation model. The use of sparse representation aims to address the challenge of catastrophic forgetting in a reinforcement learning (RL) setting, where data generation constantly shifts. The architecture employed for model-based reinforcement learning is Dyna. The proposed method involves learning nonlinear sparse features and building a model based on this sparse representation. To enhance computational efficiency, a method for updating model weights using sparse representation is presented. Empirical experiments are included to demonstrate the effectiveness of this approach.

**Strengths:**

1. The paper targets an important topic of learning a model in RL

2. I do not see many related works of building sparse representation-based models.

**Weaknesses:**

I will list below main weaknesses for improvements, centred around the main contribution of the paper.

1. Is there any reason for why the particular sparse representation learning method is chosen? Furthermore, in the experiments part, FTA should also be compared as a baseline. It is unclear why you compare it in a supervised learning setting but omit in a RL setting. The performance on a SL setting does not invlidate/validate another. As an empirical paper, I think a rigorous comparison is necessary.

2. Could you clarify do you update both your model and representation every environment time step?

3. There is a critical weakness in the paper: the paper claims to develop a sparse representation-based approach for model learning, but it is not justified the reported benefits come from the use of the sparse representation for policy learning or for model learning. Note that the former has been extensive studied. in general, a full replay method should be the best in mitigating catastrophic forgetting, but the empirical results reported that the proposed algorithm can sometimes even outperform full replay. That raises a natural question that the benefit mainly comes from the policy learning part by using sparse representation, rather than the proposed model learning part.

4. other issues.

Alg 1. it is better to be specific, use the title Dyna architecture, rather than MBRL, as there are numerous MBRL algorithms and not everyone is as Alg 1 described.

Alg 2, line 4 & 5: shouldn’t it be outer product? Please specify the dimension of the matrices capital Phi and letter phi. This is nontrivial as it affects the understanding of the algorithm.

The term “world model” might intrigue the readers to see much more challenging tasks than the paper presented, this can be seen by other papers using such terms. It is better to rephrase it to be more precise.

**Questions:**

see above.

---

> ### Author Response · Authors · 2023-11-15
> **Response to Reviewer 2Y3F (1/2)**
>
> Thank you very much for recognizing that the problem we attempt to tackle (learning models online) is important in RL, as well as your constructive feedback and suggestions. We have updated the paper with text changes highlighted in yellow, and elaborate on each of the questions below. Note that the points below do not match those in the original questions due to reorganization, but they cover all questions.
>
> **1. Why is the particular sparse representation chosen?**
>
> We adopt linear models to have guaranteed online learning performance without forgetting. However, linear models cannot fit complex functions. Adding a nonlinear random feature turns linear models into universal approximators [1], but this usually requires a very wide layer to have good capacity. We make our high-dimensional representation sparse to achieve efficient incremental update. In addition, we also compare our sparse representation with other encoding methods and we show favorable results (Table 1). The idea of our sparse update is inspired by the model mixer in [2].
>
> We have updated the paper on page 6 with a “concluding remark” to make this point clearer.
>
> **2. Why is FTA not included in the experiments?**
>
> Fuzzy Tiling Activation (FTA) [3] proposes an effective way to reduce catastrophic forgetting of neural networks in online continual learning problems, by learning sparse activations inside the NNs. In Section 4.2, we only borrow their nonstationary online learning setting (the Piecewise Random Walk) and show our method is more capable than NNs of learning temporally correlated data. However, we did not include experiments to compare FTA and our method in either SL or RL settings, for the reasons below.
>
> In SL, from Figure 3.c of [3], we can see FTA slightly suffers from nonstationarity if the data is highly correlated. The reason could be that FTA only mitigates the catastrophic forgetting issue in NNs, but it does not fully solve the problem. Therefore, we explore a more assured pathway in this work, by which a Follow-the-Leader solution can be obtained online efficiently, aiming to **eliminate** catastrophic forgetting. As Figure 3 in our paper suggests, Losse-FTL can attain very low error for all levels of correlation, showing its non-forgetting property.
>
> In RL, we have attempted to integrate the official FTA codes into NN-based world models, but we found it hard to make it work well (perhaps due to hyper-parameter tuning, such as the tiling dimension $k$ and the sparsity factor $\eta$, or where the FTA layer should be applied). Nevertheless, we compared our method with other representative continual learning techniques, including Synaptic Intelligence and Coreset, to show its effectiveness.
>
> **3. Do you update both your model and representation every environment time step?**
>
> Our high-dimensional sparse representation is not learnable with a prefixed random projection matrix. Thus we only update the model (the linear weights) but not the representation.
>
> For all RL experiments, we keep the update interval as $25$ time steps for discrete control tasks and $250$ for continuous control tasks to accelerate the experiments. However, we note that our updates are much more efficient compared to NN-based ones with replay (referring to Figure 5) because the latter needs to train until convergence over accumulated experiences.

---

> ### Author Response · Authors · 2023-11-15
> **Response to Reviewer 2Y3F (2/2)**
>
> **4. Does the benefit come from policy learning or model learning?**
>
> Thanks for this insightful question. We agree that sparse representation helps continual policy learning [3,4,5], as it could potentially reduce the catastrophic interference in NNs. However, in this work, we utilize the high-dimensional sparse representation **only during model learning** for Losse-FTL, while keeping policy/value learning the same for apple-to-apple comparison. Therefore, we believe the benefit indeed comes from model learning; better model learning ensures a better quality of synthetic transitions to be used for policy/value learning.
>
> **5. Why does the proposed algorithm sometimes even outperform the Full-replay baseline?**
>
> We agree that theoretically, Full-replay should be as good as ours in eliminating forgetting. We hypothesize that the reason for our method outperforming Full-replay in some tasks is due to the criterion we adopt for “convergence” – at which point we stop the NN training, while ours is solved incrementally in closed form. Apart from the convergence criterion, there is more brittleness in NN training in this case. For example, not resetting the weights may make NNs overfit early data; but when to reset is hard to know because there are no explicit task boundaries; even if we know when, resetting and training from scratch is even more time-consuming and impractical for lifelong agents.
>
> We have updated the paper on page 8 about “Full-replay” to clarify the experimental settings.
>
> **6. Other issues.**
>
> Thanks for pointing out the issue of Algorithm 1 and the need for better notations. We have updated them directly in the text. To be more specific on Dyna architecture, we remove the original Algorithm 1 and use a diagram of Dyna in Figure 2 to give a more intuitive illustration in Section 2.2.
>
> It also took us some time to consider the term “world model” in this context. We agree that it can refer to tasks that are more challenging, as envisioned in [6]. We hope to retain it for several reasons. (1) Simply using “model” is common in RL contexts, but we feel it might be confusing since “model” is frequently referred to with different contexts, e.g. linear model. In the title, especially, without some hints about RL, people may not recognize “model” refers to the world model. (2) Other terms like “dynamics model” only describe the dynamics part but not the reward part. We also thought of the “environment model”, however, it is not very commonly seen in literature. In the paper, we also tried to put the term “world model” in the context of MBRL, to avoid misunderstanding. Please let us know if there are more suitable terms.
>
>
> ---
> Thank you again for your detailed feedback and suggestions, and we hope our responses and new experiments were able to address any remaining concerns. Please do let us know if you have any further questions as well as what would be expected for score improvement.
>
>
> ### References
>
> [1] Huang, G. B., Chen, L., & Siew, C. K. (2006). Universal approximation using incremental constructive feedforward networks with random hidden nodes. IEEE Trans. Neural Networks.\
> [2] Knoll, B., & de Freitas, N. (2012). A machine learning perspective on predictive coding with PAQ8. In 2012 Data Compression Conference.\
> [3] Pan, Y., Banman, K., & White, M. (2021). Fuzzy tiling activations: A simple approach to learning sparse representations online. In ICLR.\
> [4] Liu, V., Kumaraswamy, R., Le, L., & White, M. (2019). The utility of sparse representations for control in reinforcement learning. In AAAI.\
> [5] Lan, Q., & Mahmood, A. R. (2023). Elephant Neural Networks: Born to Be a Continual Learner. In ICML Workshop on High-dimensional Learning Dynamics.\
> [6] LeCun, Y. (2022). A path towards autonomous machine intelligence version 0.9. 2, 2022-06-27. Open Review, 62.

---

> ### Author Response · Authors · 2023-11-20
> **Looking forward to further feedback**
>
> Dear Reviewer 2Y3F,
>
> Thank you again for your valuable comments and suggestions, which are very helpful to us. We have posted responses to the proposed concerns.
>
> We understand that this is quite a busy period, so we sincerely appreciate it if you could take some time to return further feedback on whether our responses resolve your concerns. If there are any other comments, we will try our best to address them.
>
> Best,
>
> The Authors

---

> > ### Comment · Reviewer_2Y3F · 2023-11-21
> >
> > Thank you for your response. I updated my rating.

---

> > > ### Author Response · Authors · 2023-11-23
> > >
> > > Thank you very much for the score improvement and your constructive feedback. We will further polish the paper in the final revision. Thank you!

---

### Meta-Review · Area_Chair_GF5s · 2023-12-12

**Metareview:**

This paper develops a model-based reinforcement learning (MBRL) method. It seeks to develop a method for learning world model with efficient incremental updates. Specifically, the world model is a linear regression supported by nonlinear random features (based on locality sensitive encoding that is sparse in nature). All the reviewers agree that the paper addresses an important problem in MBRL, and that the work makes an interesting contribution. The reviewers also agree that the concerns raised during review have been sufficiently addressed. The authors are encouraged to take all the feedback into account when revising their paper for final publication.

**Justification For Why Not Higher Score:**

There are a few weak side of the paper such as not comparing to the state-of-the-art baselines. Although it may not be the major issue for the paper, it still makes it less strong for a higher tier such as spotlight or oral.

**Justification For Why Not Lower Score:**

This paper makes meaningful contribution to an important problem in MBRL problem. It is worth publication in the venue.

---

### Decision · Program_Chairs · 2024-01-16

Accept (poster)